# Recent Insight into the Role of Sphingosine-1-Phosphate Lyase in Neurodegeneration

**DOI:** 10.3390/ijms24076180

**Published:** 2023-03-24

**Authors:** Iga Wieczorek, Robert Piotr Strosznajder

**Affiliations:** Laboratory of Preclinical Research and Environmental Agents, Mossakowski Medical Research Institute, Polish Academy of Sciences, Pawińskiego 5 St., 02-106 Warsaw, Poland

**Keywords:** sphingosine-1-phospahte (S1P), S1P lyase (SPL), neurodegeneration, neuroinflammation, SPL inhibitors

## Abstract

Sphingosine-1-phosphate lyase (SPL) is a pyridoxal 5′-phosphate-dependent enzyme involved in the irreversible degradation of sphingosine-1-phosphate (S1P)—a bioactive sphingolipid that modulates a broad range of biological processes (cell proliferation, migration, differentiation and survival; mitochondrial functioning; and gene expression). Although SPL activity leads to a decrease in the available pool of S1P in the cell, at the same time, hexadecenal and phosphoethanolamine, compounds with potential biological activity, are generated. The increased expression and/or activity of SPL, and hence the imbalance between S1P and the end products of its cleavage, were demonstrated in several pathological states. On the other hand, loss-of-function mutations in the SPL encoding gene are a cause of severe developmental impairments. Recently, special attention has been paid to neurodegenerative diseases as the most common pathologies of the nervous system. This review summarizes the current findings concerning the role of SPL in the nervous system with an emphasis on neurodegeneration. Moreover, it briefly discusses pharmacological compounds directed to inhibit its activity.

## 1. Introduction

Sphingolipids, a family of lipids containing an 18-carbon-chain amino alcohol sphingosine (Sph) as a backbone, are crucial components of eukaryotic membranes and essential signaling molecules with different effects on cell fate. On the one hand, ceramides (Cer), central elements in sphingolipid turnover (they are a substrate for the synthesis of the other sphingolipids), promote cell senescence, cell-cycle arrest and apoptosis [1]. On the other hand, sphingosine-1-phosphate (S1P), a phosphate derivative of Sph, a product of Cer catabolism, exhibits prosurvival activity [2]. S1P exerts both intra- and extracellular activity. Extracellularly, it acts via specific G protein-coupled receptors (S1PR1-S1PR5) and, this way, mediates proliferation and cell migration, differentiation and survival [3,4,5]. Intracellularly, S1P regulates mitochondrial function and endoplasmic reticulum stress and modulates gene expression [6,7,8,9].

Since S1P is involved in multiple cellular processes, its level has to be tightly controlled, which is the responsibility of highly specialized enzymes (Figure 1). The balance between S1P and Sph is maintained by two isoforms of sphingosine kinase (SphK), cytosolic SphK1 and, localized in the nucleus, endoplasmic reticulum (ER) and mitochondria SphK2 [10], which catalyze the phosphorylation of Sph to S1P. In turn, sphingosine-1-phosphate phosphatases (S1PPs), localized in ER, mediate the reversible dephosphorylation of S1P to Sph. Nonetheless, the crucial reaction of S1P metabolism is governed by sphingosine-1-phosphate lyase (SPL), which irreversibly cleaves S1P between carbon atoms C2 and C3 [11] to hexadecenal (HE) and phosphoethanolamine (P-Etn), thereby reducing the available pool of S1P in the cell [12].

SPL (EC 4.1.2.27) is an intracellular membrane protein localized almost equally in smooth and rough ER (although a recent study of the group of Natarajan revealed its presence in a nuclear fraction from lung epithelial cells [13]), which belongs to the superfamily of pyridoxal 5′-phosphate (PLP)-dependent enzymes [14]. The human SPL gene (*SGPL1*), localized on chromosome 10q22, encodes a protein consisting of 568 amino acids, with an approximated molecular mass of 63.5 kDa, while *SGPL1* cDNA consists of 15 exons [15]. Human SPL displays 84% amino acid identity and 91% similarity to the mouse orthologue [16]. In its structure, we can distinguish an N-terminal lumenal domain, a transmembrane segment and a soluble, exposed to the cytosol, PLP-binding domain (with lysine residue (Lys-353) as an active site that forms an internal Schiff base with PLP [17]) responsible for the catalytic activity [18]. SPL acts as a dimer with two catalytic sites to which each promoter provides residues simultaneously [18,19]. The distribution of SPL in a body is not uniform and varies depending on the tissue. In rodents, the highest level of SPL was observed in the small intestine, thymus, colon and spleen. Moderate enzyme levels were noticed in the liver, kidney, lung, stomach and testis. Brain (apart from olfactory mucosal epithelium), heart and skeletal muscle express barely detectable amounts, whereas SPL is completely absent in platelets and erythrocytes [14,17]. In humans, the highest SPL protein expression was observed in the brain (basal ganglia, hippocampus, brain cortex and cerebellum) and female organs (placenta, endometrium, fallopian tubes and ovaries). Moderate enzyme levels of the enzyme were identified in thyroid glands, respiratory and digestive system (apart from the proximal digestive tract), kidney and urinary bladder, male organs (testis, prostate), bone marrow and such lymph organs as lymph nodes and tonsils. Salivary glands, heart muscle and skin express low SPL levels, while in eyes and connective and soft tissues, SPL is absent [20].

Ubiquitous expression of SPL proves its important role in the body’s functioning, and thus even small changes in its level/activity can lead to severe impairments. Constitutive (systemic) knockout of the SPL gene (*Sgpl1*) in mice produced developmental impairment (kidney defects, embryonic hemorrhaging, lower level of red blood cells, reduced size and weight) and early (8 weeks after birth) postnatal lethality [21]. In humans, at least 16 different variants of *SGPL1* resulting in a lack of or reduced protein expression and/or activity were identified, which had severe health consequences [22]. Loss-of-function mutations in the SPL encoding gene are associated with primary adrenal insufficiency (PAI) and steroid-resistant nephrotic syndrome (SRNS). Mutation carriers also experience skin abnormalities (e.g., ichthyosis), genital defects (hypogonadism, cryptorchidism), hypothyroidism, neurodevelopmental delay and severe lymphopenia [22,23,24,25,26,27,28]. On the cellular level, *SGPL1* mutations contribute to mitochondrial dysfunctions, including altered morphology (hyper-fused and elongated or fragmented organelle depending on the mutation); disturbed mitochondrial dynamics; and oxidative phosphorylation accompanied by enhanced levels of S1P, Sph, sphingomyelin and Cer species [29]. Lack of SPL is also connected to impaired lipid metabolism manifested in: disruption of intracellular cholesterol trafficking; higher levels of cholesterol, phospholipids and triglycerides in serum; and increased lipid accumulation in the liver [30,31]. It is noteworthy that SPL plays a significant role in carcinogenesis. Decreased activity and/or expression of SPL was observed in human prostate cancer [32] and human colon cancer [33]. Moreover, SPL loss may be one reason for cell resistance to chemotherapy [32,34].

It remains to be elucidated if the above-mentioned deleterious effects of SPL deficit are associated with the lack of HE and P-Etn, end products of SPL-mediated reaction, which are thought to exert biological activity. Human (HEK293T, HeLa) and murine (NIH3T3) cell lines treated with exogenous 2-trans-hexadecenal displayed cytoskeletal reorganization, cell detachment and apoptosis [35]. Observed effects were a consequence of c-Jun terminal kinase (JNK) activation, which was dependent on a mixed lineage kinase 3 (MLK3) and the reactive oxygen species (ROS) generation. Activation of this signaling pathway resulted in c-Jun phosphorylation, modulation of Bcl-2 family member proteins (Bax activation, Bid cleavage and enhanced Bim translocation into mitochondria) and cytochrome c release [35]. It was shown that HE activated Bax by its direct covalent lipidation at C126 [36]. Moreover, HE was proven to form adducts with DNA [37] and proteins and glutathione conjugates [38]. Intriguingly, HE was also formed in the non-enzymatic, free-radical-induced degradation of S1P [39]. Recently, it was unveiled that HE produced by the nuclear isoform of SPL stimulated histone acetylation: histone 3 at lysine 9 (H3K9) and histone 4 at lysine 8 (H4K8) in lung epithelial nuclear fraction [13]. Moreover, it modulated the activity of histone deacetylases 1 and 2 (HDAC1 and HDAC2, respectively), for which the ability of HE to form adducts with HDAC1 is most likely responsible [13]. As HE regulates (in a concentration-dependent manner) ROS production, mitochondrial potential and apoptosis in polymorphonuclear leukocytes (PMNLs), cells considered the first line of defense against infections, it may strongly affect the functioning of the immune system [40]. In turn, P-Etn may participate in the synthesis of phosphatidylethanolamine (PE) in the cytidine diphosphate (CDP) ethanolamine branch of the Kennedy pathway [41]. Furthermore, P-Etn generated by SPL is crucial to the viability and differentiation of Leishmania major [42], and the viability of Trypanosoma brucei [11].

Along with the progress of research on the importance of SPL in physiological and various pathological conditions, the enhanced interest in the role of S1P, its metabolism and S1P-dependent signaling in the nervous system and related diseases has been observed. In this review, we want to discuss the recent findings concerning the role of SPL in the nervous system, paying particular attention to neurodegeneration as the most common nervous system pathology. Finally, we want to summarize the up-to-date knowledge about inhibitors of SPL, which are seen as promising compounds in the pharmacotherapy of various diseases.

## 2. SPL in the Nervous System

S1P and S1P-dependent signaling play an important role in the functioning of the nervous system, as shown by the presence of S1PR1-3 already at the early stage of nervous system development [43]. S1P is involved in the proliferation, differentiation and survival of neurons and neural progenitors [44,45]. S1P participates in neurogenesis, both under physiological conditions and after brain damage [46,47], and mediates nerve growth factor-(NGF)-induced neurite outgrowth and increased excitability in sensory neurons [48,49]. S1P is necessary for neural tube closure, as SphK1^−/−^SphK2^−/−^ mice unable to produce S1P showed neural tube defects [50]. It also serves as a chemoattractant for neural progenitor cells, which stimulates their migration to diseased parts of the central nervous system (CNS) [51]. S1P has also been demonstrated to induce the gene expression of neurotrophic factors, such as brain-derived neurotrophic factor (BDNF), heparin-binding EGF-like growth factor (HB-EGF), leukemia inhibitory factor (LIF) and platelet-derived growth factor B (PDGFB) in both murine and human astrocytes, for which simultaneous activation of S1PR1 and S1PR2 is necessary. It also confers indirect protection of hippocampal neurons against N-methyl-d-aspartate (NMDA)-induced excitotoxicity when neurons are cocultured with glial cells [52]. However, prolonged exposure to a high concentration of S1P triggered Ca^2+^ release from the inositol 1,4,5-trisphosphate (IP3)-sensitive internal stores and led to apoptosis in cultured hippocampal neurons [53]. A more recent study demonstrated that S1P-associated neuronal death was mediated by Ca^2+^-dependent cysteine protease calpain [54]. The fact that an excess of S1P, considered a prosurvival molecule, provokes cell death emphasizes the importance of S1P-degrading enzymes, including SPL, in maintaining the proper functioning of the nervous system.

Some of the aforementioned mutations in the gene coding for SPL, which are responsible for PAI and SRNS, also account for neurological impairments in their carriers, such as sensorineural deafness, ataxia, muscular hypotonia and severe brain malformation (microcephaly, cerebellar hypoplasia) [24,25,26], what corroborates the crucial role of SPL in early stages of neurodevelopment. The partial degradation of SPL as a result of mutations in *SGPL1*, concomitant with enhanced plasma levels of S1P and the Sph/sphinganine ratio, were observed in patients affected by autosomal recessive forms of Charcot-Marie-Tooth disease—a group of hereditary motor and sensory neuropathies [55]. These mutations were responsible for atypical symptoms of the disease, such as acute/subacute onset, unilateral motor deficit and episodes of mononeuropathy with a tendency for improvement. An important role of SPL in the proper functioning of peripheral nerves confirmed studies on *Drosophila melanogaster* (*D. melanogaster*). Flies with neuron-specific downregulation of *Sply* (*Drosophila* orthologue of *SGPL1*) manifested defective neuromuscular junction morphology and progressive degeneration of the chemosensory neurons innervating the wing margin bristles [55].

SPL appears to play an important role in synaptic transmission, as a selective lack of SPL in a murine brain led to a considerable decrease in the level of presynaptic proteins (markers Bassoon, synapsin-1, syntaxin-1a and synaptobrevin-2), and a reduction in number and density of synaptic vesicles in the hippocampus [56]. Furthermore, SPL deficit may be linked to neuronal death [57]. Incubation of SPL-deficient neurons with either exogenous S1P or Sph led to comparable S1P accumulation in the cells. Nevertheless, only treatment with S1P decreased cell viability. The underlying mechanism is S1P-receptor independent but related to dephosphorylation of exogenous S1P to Sph and rephosphorylation conducted by SphK2, whereas Sph was phosphorylated by SphK1, giving a non-toxic product [57,58].

Numerous investigations indicate autophagy as a key process for the survival of neural cells [59,60,61,62]. Searching for the molecular bases of autophagy indicates S1P as a probable player in regulating this process. Overexpression of SphK1 promoted the generation of pre-autophagosomal structures. In contrast, expression of dominant-negative SphK1 (unable to phosphorylate Sph to S1P) inhibited autophagosome formation in cultured primary neurons, indicating that SphK1 is involved in the biogenesis of autophagosomes. Unlike SphK1, which enhanced autophagic flux, S1PP and SPL inhibited this process [63]. Another study, however, seems to contradict the inhibitory role of SPL in autophagy. In mice with neural-specific ablation of SPL in the brain, it was observed that disturbed autophagy is caused by a significant decrease in the level of brain PE, a product of P-Etn transformations [64]. Numerous studies show that PE plays an important role in the formation of autophagosome [65,66]. Furthermore, disturbed autophagy was associated with enhanced neuronal accumulation of proteins susceptible to aggregation, such as APP and its derivatives and α-synuclein [64,67], while PE addition restrained this process [64], indicating a possible relationship between decreased SPL activity and neurodegenerative proteinopathies.

## 3. SPL in Neurodegenerative Diseases

Neurodegeneration is characterized by progressive loss of functional activity and death of neurons, which leads to a wide range of diseases affecting the central or peripheral nervous system. A large body of evidence highlighted the importance of sphingolipids, including S1P, in different neurodegenerative diseases like Alzheimer’s disease (AD) [68,69,70,71], Parkinson’s disease (PD) [72,73,74], Huntington’s disease (HD) [75,76,77,78,79], amyotrophic lateral sclerosis (ALS) [80,81,82] and spinal muscular atrophy (SMA) [83,84], although the role of SPL is a subject of study only in few of them (Figure 2).

### 3.1. Alzheimer’s Disease

Alzheimer’s disease (AD) is a neurodegenerative disease and the most common cause of dementia worldwide. Clinical manifestations of the disease, such as progressive loss of memory and impaired cognitive and social functions [85,86], reflect alterations in the cortex and hippocampus, brain structures particularly affected in AD [87]. These changes are associated with the toxicity of the hyperphosphorylated tau protein, which aggregates as intracellular neurofibrillary tangles (NFTs), and different forms (soluble monomers, oligomers and insoluble fibrils and plaques) of amyloid β peptide (Aβ). The latter occurs as a result of aberrant cleavage of APP, a transmembrane protein responsible for the regulation of synaptic formation and function. In this proamyloidogenic pathway, β-secretase (beta-site APP cleaving enzyme 1, BACE1) acts as a rate-limiting enzyme [88]. SPL seems to be of great importance in APP processing and, consequently, in Aβ-associated pathology such as AD. Fibroblasts from SPL knockout (SPL-KO) mice showed impaired APP metabolism [89]. Functional lack of SPL connected with high amounts of intracellular S1P caused the accumulation of full-length APP and its potentially amyloidogenic C-terminal fragments (CTFs) due to their impaired degradation in lysosomes. Moreover, the decrease of γ-secretase activity, an enzyme participating in a second step of APP cleavage, was observed [89]. In an experiment on mouse primary neurons, pharmacological inhibition of SPL by THI evoked increased Aβ secretion [90]. On the contrary, overexpression of SPL in N2a neuroblastoma cells resulted in highly reduced levels of βCTFs and Aβ secretion. The possible mechanism is that SPL, regulating the S1P level, may indirectly modulate BACE1 activity, as a decreased S1P level due to inhibition or knockdown of SphKs led to reduced activity of this protease [90].

In contrast to the aforementioned studies, clinical observation showed a positive correlation between Aβ deposits in the entorhinal cortex and increased protein levels of SPL. At the same time, decreased protein levels of SphK1 were observed [91], which suggests that a low level of S1P may be related to impaired Aβ clearance. However, another post-mortem brain examination revealed that loss of S1P occurred already at the early stage of AD, and there is no correlation between reduced S1P level in the hippocampus and Aβ level [92]. Although the above-mentioned studies have demonstrated that SPL may regulate APP metabolism, there are also experiments indicating the opposite, i.e., increased SPL level as a result of enhanced APP or Aβ level. An advantage of the S1P degrading enzyme over S1P kinases was observed in the brain cortex of five familial Alzheimer’s disease (5xFAD) transgenic mice aged 8 and 14 months but not in 3-month-old individuals [93]. Furthermore, disturbed S1P metabolism was accompanied by increased S1PR1 level and enhanced activity of its downstream Akt/mTor/Tau signaling pathway in aging animals. In all groups, an increased level of APP was observed [93]. A higher protein level of SPL, together with an enhanced protein level of S1PP and a decreased level of SphK2, was also demonstrated in a model of memory deficit induced by intrahippocampal Aβ injection [94].

SPL may also be associated with AD pathology by influencing epigenetic modifications of gene transcription, such as acetylation and deacetylation of histone proteins. Previous studies on AD animal models, as well as post-mortem brains of AD patients, showed the imbalance between these two processes [95,96,97]. Experiments on hippocampal and cortical slices from SGPL1^fl/fl/Nes^ mice revealed a marked increase in tau phosphorylation at sites relevant to AD accompanied by enhanced acetylation of H3K9 and histone 4 at lysine 5 (H4K5) [98]. In turn, experiments on primary cultured neurons and astrocytes from SGPL1^fl/fl/Nes^ mice showed cell-specific changes, namely hyperphosphorylated tau only in SPL-deficient neurons and elevated histone acetylation only in SPL-deficient astrocytes. All alterations can be reversed by Ca^2+^ chelation with BAPTA-AM [98]. Increased H3K9 acetylation, but not H4K5, associated with decreased activity of HDACs was also observed in SPL-deficient mouse embryonic fibroblasts (MEFs). Moreover, reduced activity and/or expression of HDAC1 and HDAC2 contributed to the increased basal intracellular concentration of Ca^2+^ [99].

### 3.2. Parkinson’s Disease

Another common neurodegenerative disease is Parkinson’s disease (PD), characterized by selective loss of neurons in the substantia nigra. Current knowledge about the role of SPL in PD remains limited. Studies on the in vitro PD model, namely the human neuroblastoma cell line (SH-SY5Y) treated with a neurotoxin 1-methyl-4-phenylpyridinium (MPP^+^), showed a significant increase in mRNA level of SPL and SphK2 together with decreased gene expression of SphK1 [100].

### 3.3. Huntington’s Disease

Huntington’s disease (HD) is an autosomal dominant inherited neurodegenerative disease caused by mutations in the huntingtin (Htt)-encoding gene (*HTT*). An expansion of CAG trinucleotide repeats (more than 36), which encode polyglutamine (polyQ) tract, leads to the mutant Htt with toxic properties. The most affected parts of the brain are the striatum and cortex [77,101]. A large body of evidence from studies on different preclinical models revealed defective expression of enzymes participating in the de novo synthesis of sphingolipids. Decreased expression of genes coding for long chain base subunit 1 of serine palmitoyltransferase (SPTLC1) and ceramide synthase 1 (CerS1) was figured out in the striatum and cortex of R6/2 mice (the first transgenic mouse model of HD), which was subsequently reflected in a significant reduction in dihydrosphingosine (dhSph), dihydrosphingosine-1-phosphate (dhS1P) and dihydroceramide C18:0 (dhCer) in both studied brain regions [76]. The post-mortem examination of the brain tissue of HD patients revealed markedly enhanced levels of SPL in the striatum and brain cortex. At the same time, the level of SphK1 in the striatum was significantly decreased in comparison to age-and-gender-matched healthy control and no change was observed in SphK2 level in any examined structures [77]. Clinical outcomes correspond to results from animal studies. The increased protein level of SPL, along with diminished protein level of SphK1 in the striatum, was noted in YAC128 mice—another transgenic model of HD, while in R6/2 mice, similar changes were also seen in cortical tissue. Moreover, changes in SPL in both analyzed structures were already present in young animals of this model representing the early stage of the disease. Additionally, mass spectral data showed a defective level of S1P (striatum, cortex), Sph (striatum) and different species of Cer (C24:0 in striatum and C20:0, C22:0, C24:0 and C24:1 in cortex) [77].

### 3.4. Amyotrophic Lateral Sclerosis

Unlike AD, PD and HD, which cause degeneration of the CNS, amyotrophic lateral sclerosis (ALS) affects, in particular, lower and upper motor neurons. During the course of the disease, patients develop muscle weakness and spasticity, speech and swallowing difficulties, fasciculations and changes in reflexes which, finally, lead to death due to severe respiratory paralysis. The etiology of the majority of the cases remains unknown (it is a so-called sporadic form of the disease), while in only 5–10% of patients the disease is associated with mutations (predominantly autosomal dominant) in genes coding for superoxide dismutase 1 (SOD1), TAR DNA-binding protein 43 (TDP-43) and RNA-binding protein FUS/TLS (FUS) [80]. Besides muscular system impairments, a characteristic feature of the disease, connected to mutations in the *SOD1* gene, is disrupted lipid metabolism, including sphingolipids [102,103]. Previous studies reported significant changes in levels of different species of glycosphingolipids accompanied by alterations in enzymes implicated in their metabolism in muscle and/or spinal cord of ALS patients and in animal models of the disease (SOD1^G93A^ and SOD1^G86R^ mice) [104,105]. Mutations in the gene coding for serine palmitoyltransferase (SPT)—a rate-limiting enzyme of ceramide synthesis, manifesting as increased SPT activity and subsequent unrestrained sphingoid base generation, are associated with childhood-onset ALS [82]. The mRNA level of SPL in the spinal cord of female SOD1^G86R^ mice showed an increasing trend compared to control animals, but the change was not statistically significant [80]. In turn, studies carried out on another ALS animal model, transgenic FUS(1-359) mice, demonstrated increased gene expression of SPL in the spinal cord together with significantly decreased mRNA level of sphingosine-1-phosphate phosphatase 2 (S1PP2), thus clearly indicating the advantage of irreversible S1P degradation over reversible dephosphorylation [81].

## 4. SPL in Neuroinflammation

Inflammation is a specific reaction of the immune system to harmful factors such as pathogens, toxic substances, damaged cells or irradiation. Although its primary role is to initiate the chain of molecular and cellular events to restore disturbed homeostasis, prolonged inflammation has detrimental effects on body functioning. Inflammatory response within the brain or spinal cord, called neuroinflammation, is considered a crucial component of neurodegenerative diseases [106,107,108,109,110]; hence knowledge about the role of SPL in neuroinflammation may help understand the molecular basis of neurodegeneration.

In the experimental immune encephalomyelitis (EAE), a rodent model of multiple sclerosis (MS), decreased neuroinflammation connected to SPL partial deficiency or pharmacological inhibition of the enzyme was demonstrated by reduced T cell (CD4^+^ and CD8^+^) migration into CNS [19,111]. In contrast, SGPL1^fl/fl/Nes^ mice demonstrated S1P accumulation in the brain, followed by enhanced activity of microglia and disturbed microglial autophagy [112]. Intrahippocampal administration of Aβ led to an increase in proinflammatory markers: tumor necrosis factor-alpha (TNF-α) and cyclooxygenase-2 in the brain, accompanied by enhanced protein level of SPL. However, unilateral intracerebroventricular injection of LPS resulting in the same neuroinflammatory effect caused, surprisingly, a decrease in SPL protein level [94]. Knockdown of the SPL gene in the human cerebral microvascular endothelial cell line (HCMEC/D3) led to the increase in expression of an important neutrophil chemotactic factor, namely interleukin 8 (IL-8), under inflammatory conditions (treatment with TNF-α) [113], which normally accompany disrupted integrity of blood-brain barrier (BBB) [114]. Interestingly, a significant decrease in gene expression of proinflammatory interleukin 6 (IL-6), vascular cell adhesion molecule 1 (VCAM-1) and monocyte chemoattractant protein 1 (MCP-1) was observed [113].

## 5. SPL Inhibitors

As shown above, SPL plays a significant role in a variety of conditions. The fact that, in most cases, increased expression/activity of SPL accounts for final detrimental effects set the direction of looking for effective inhibitors of this enzyme (Table 1).

### 5.1. Sphingosine Analogues

#### 5.1.1. Fingolimod

Fingolimod (FTY720), a synthetic analogue of Sph, was discovered as a result of modification of myriocin, a fungal metabolite with immunosuppressive properties isolated from the fungus Isaria sinclairii [141,142,143]. Radical simplification of myriocin structure led to lower toxicity and enhanced immunosuppressive activity of FTY720—the latter probably due to the changes in the mechanism of action (unlike myriocin, FTY720 does not inhibit SPT) [142,144]. FTY720 was initially considered a drug for antirejection therapy in combination with well-established medicine like cyclosporin A after renal transplantation [145,146,147,148]. Currently, it is approved for treating the relapsing-remitting form of MS as the first oral drug for this disease [149].

FTY720 is a prodrug, and it has to undergo in vivo phosphorylation (mostly by SphK2 in the liver [150,151]) to its active state, namely FTY720-phosphate (p-FTY720). p-FTY720 binds four of five S1P receptors (S1P1,3-5), which are present on a broad spectrum of cells among others on T cells, in this way preventing their egress from thymus and secondary lymphoid tissues to peripheral circulation and subsequently leading to significant peripheral lymphopenia [152,153,154,155]. Numerous studies demonstrated that S1PR1 plays a key role in the aforementioned process [152,156,157,158,159]. A more detailed explanation of the fingolimod lymphopenic effect is that it causes the downregulation of S1PR1 on lymphocytes [152] and endothelial cells [160]. Fingolimod not only impacts T cell trafficking but also affects their activation, which occurs in a receptor-independent manner [161,162,163]. It also should be mentioned that FTY720 can act in CNS. Once it crosses the BBB, it undergoes phosphorylation (due to the presence of SphK2 in CNS) and binds S1PRs distributed on both neurons and glial cells [164].

In addition to the aforementioned mechanisms of action of FTY720, it was demonstrated, both in in vitro and in vivo studies, the inhibitory effect of fingolimod on SPL [115]. In vitro experiments on HEK293 cells demonstrated dose-dependent inhibition of SPL by unphosphorylated FTY720, starting at a concentration of 300 nM and with a peak at a concentration of 30 µM. Conversely, treatment with p-FTY720 failed to suppress SPL activity (it showed inhibitory properties only at 30 µM and its potency was lower than that of FTY720 at this dose) [115], which was corroborated by the results from an experiment by Berdyshev et al. on liver microsomal fraction treated with 30 µM p-FTY720 [116]. The latter study also proved (reaction conducted on total tissue homogenate and the microsomal protein fraction) that unphosphorylated fingolimod suppressed SPL with an IC_50_ of 52.4 µM [116]. CD68^+^ antigen-presenting cells from human monocytes (MoAPCs) expressing high levels of SPL were shown to degrade S1P into HE—process strongly inhibited by FTY720 [117]. In vivo studies on 4-week-old female FVB mice, which received 1 mg/kg FTY720 intraperitoneally, demonstrated inhibition of SPL activity already 12 h after treatment (maximal effect at 36 h), and this occurred along with lymphopenia. At the same time, no significant influence of FTY720 was observed on SPL gene expression or protein level [115]. As previously mentioned, a congenital lack of SPL activity is connected to severe adrenal insufficiency. However, modification of SPL activity during long-term treatment with fingolimod did not affect adrenal function [165].

#### 5.1.2. Other Sphingosine Analogues

There are also other analogues of Sph which were demonstrated to inhibit SPL. The first mentioned in the literature is 1-desoxysphinganine 1-phosphonate, a competitive inhibitor with Ki = 5 μmol/L. Studies in vivo showed high compound toxicity, causing immediate death after intravenous administration and hemolytic properties already at low concentration [139]. Another example is 2-vinyl dihydrosphingosine-1-phosphate (2VS1P), a mixture of two diastereomers, which demonstrated an IC_50_ of 2.4 μM [140]. Although this compound was forgotten for a long time, nowadays, the synthesis of vinyl derivatives of sphingolipids is again drawing the scientists’ attention [141]. Until now, the last Sph analogue that may act as an SPL inhibitor was N-[(1R,2S)-2-hydroxy-1-hydroxymethyl-2-(2-tridecyl-1-cyclopropenyl)ethyl] octanamide (GT11)—a ceramide analogue proved to be a potent, dose-dependent (at a range of concentrations of 0.001 to 1 μM) inhibitor of dihydroceramide desaturase. Preincubation of primary cultured cerebellar neurons from 6-day-old mice with 10 μM GT11 led to a marked reduction in SPL activity without affecting mRNA level, whereas lower concentration (1 μM) had no effect on SPL [142].

### 5.2. Non-Lipidic Direct Inhibitors of SPL

High-throughput screening (HTS) made it possible to obtain compounds that are not Sph analogues but directly affect SPL activity, such as oxopyridylpyrimidine (IC_50_ = 2.1 μM). Although it was demonstrated to increase S1P level by 275% over control in the human hepatoma cell line (HepG2 cells), it failed to exert activity in vivo, possibly due to high plasma protein binding [121]. Another member of this group is (R)-6-(4-(4-Benzyl-7-chlorophthalazin-1-yl)-2-methylpiperazin-1-yl)nicotinonitrile (Compound 31). In an experiment on purified human SPL, Compound 31 bound to the active side of the enzyme with IC_50_ = 0.214 μM, while in in vitro assay on HEK293T cells, it induced a concentration-dependent increase in secreted S1P [122]. Unlike oxopyridylpyrimidine, Compound 31 demonstrated its efficacy in vivo as well. In EAE in rats, oral administration of Compound 31 led to a marked increase in S1P level in lymph nodes (without changes in the level of S1P in the brain), peripheral lymphopenia and suppression of T cell migration into CNS [19]. Despite its high efficacy, adverse effects in animals, such as platelet activation, skin irritation and the most serious: kidney toxicity [123] and bradycardia due to an increased S1P level in cardiac tissue [124], may prevent it from testing on humans.

### 5.3. Functional Antagonists

#### 5.3.1. 4-Deoxypyridoxine

4-deoxypyridoxine (DOP) is a potent antagonist of vitamin B6. As the activity of SPL is dependent on PLP (the active form of vitamin B6), DOP was reported to be a functional antagonist of SPL, i.e., it reproduces a phenotypic SPL inhibition despite the lack of activity on the isolated enzyme. Mice treated with DOP presented markedly decreased lyase activity (measured by the production of hexadecenal), which was restored by an excess of vitamin B6 [125]. Previous studies showed that DOP reduced the number of circulating lymphocytes in blood [166] and levels of proinflammatory cytokines [167]. The reduction in inflammatory response after DOP administration was also observed in experimental Crohn’s-like ileitis. Enhanced S1P level in the ileum was accompanied by diminished ileal gene expression of proinflammatory cytokines and reduced chronic ileitis [126]. Inhibition of SPL by DOP has been reported to improve the viability of isolated pancreatic islets ex vivo [127], while it reduced the level of proinflammatory cytokines in plasma and consolidated vascular endothelial barrier in a murine model of sepsis [128]. DOP also exhibits its inhibitory activity in vitro. Its administration to quiescent astrocytes resulted in a twofold increase in cellular S1P, however, without affecting cell proliferation [129]. Treatment of wild-type MEFs with DOP for 7 days resulted in significantly enhanced acetylation of histone H3K9—the same effect as in SPL-deficient cells [99].

#### 5.3.2. 2-Acetyl-4(5)-(1,2,3,4-tetrahydroxybutyl) Imidazole

Similar effects to those of DOP were observed after treatment with 2-acetyl-4(5)-(1,2,3,4-tetrahydroxybutyl) imidazole (THI) [125]—the main ingredient of the food additive Caramel Colour III [168]. THI was reported to cause the sequestration of lymphocytes in non-lymphoid organs, mainly in the liver [169]. Topical use of THI in a murine model of allergic rhinitis produced a dose-dependent reduction in the number of eosinophils and mast cells in the lamina propria of the nasal mucosa [130]. SPL inhibition by THI with subsequent increase in circulatory and cardiac levels of S1P and its endogenous analogue dhS1P resulted in improved resuscitation and survival rates after cardiac arrest in SphK1 knockout (SphK1-KO) mice; moreover, the inhibition of SPL with THI in the same animals restored reduced gene expression of S1PR2 and partially recovered decreased mRNA level of S1PR1 in the heart [131]. While cardiac ischemia-reperfusion, THI reduced infarct size and improved functional recovery in murine hearts ex vivo [132]. Elevation in plasma S1P level after THI administration in mice following acute myocardial infarction (AMI) raised mobilization of bone marrow (BM)-derived stem/progenitor cells (BMSPCs), which express genes accounting for angiogenesis and cell survival, thus contributing to decreased scar size, reduced adverse remodeling and functional cardiac recovery [133]. Apart from its peripheral effects, THI also seems to be effective in CNS. Intraperitoneal administration of THI to streptozotocin-diabetic mice reversed spatial memory impairments to a certain extent. Furthermore, it partially restored decreased Cer level in the hippocampus and normalized levels of Cer and Sph in the cerebellum [134]. In experimental models of HD, chronic administration of THI resulted in mitigation of motor deficits, activation of prosurvival signaling pathways (demonstrated among others by increased phosphorylation of kinase AKT and enhanced level of BDNF), preservation of white matter integrity and proper synaptic activity in R6/2 mice, while amelioration of locomotor function and increased lifespan in a *D. melanogaster* HD model [135].

#### 5.3.3. LX2931 and LX2932

Structural modifications of THI led to the development of two new inhibitors: (E)-1-(4-((1R, 2S, 3R)-1,2,3,4-Tetrahydroxybutyl)-1H-imidazol-2-yl)ethanone oxime (LX2931) and (1R, 2S, 3R)-1-(2-(Isoxazol-3-yl)-1H-imidazol-4-yl)butane-1,2,3,4-tetraol (LX2932), whose oral administration to mice resulted, similar to THI, in a decrease in a number of peripheral lymphocytes [136]. Prophylactic dose (100 mg/kg bw, started on day 1 before immunization) of LX2931 in a murine collagen-induced arthritis model hampered the onset of the disease manifested by a significant reduction in inflammation, erosion, exudate formation and synovial hyperplasia in tarsus. Accordingly, therapeutic dose of LX2931 and LX2932 (30 mg/kg bw, started on day 20 after immunization) gave marked amelioration of the disease as assessed by joint swelling and clinical scores [136]. Efficacy and safety data obtained from studies on animals (mice, beagles and Cynomolgus monkeys) made it possible to begin clinical trials with LX2931 in stable rheumatoid arthritis (RA) [136]. The anti-inflammatory effect of the compound was also noticed in cystic fibrosis (CF). In a mouse model of CF, oral administration of LX2931 restored diminished S1P level in the lungs leading to the correction of aberrant monocyte-derived dendritic cells (MoDC)/conventional dendritic cells (cDC) ratio, and a significant decrease in B cells and T cells counts. However, it failed to rescue disturbed fluid transport in secretory epithelia caused by a deletion in the gene coding for cystic fibrosis transmembrane conductance regulator (CFTR) [137]. In the same model of CF treated with LPS and N-formylmethionyl-leucyl-phenylalanine (fMLP), administration of LX2931 restored elevated levels of interferon-gamma (IFN-γ), interleukin 12 (IL-12), interleukin 10 (IL-10) and keratinocyte-derived cytokines (KC). Furthermore, it reduced the amount of Muc5AC—an inflammation-responsive, gel-forming type of mucin produced by goblet cells in the bronchi [137].

The literature data show that LX2931 is a subject of research also in non-inflammatory diseases. Administration of LX2931 in different concentrations to the C2C12 cell line (murine myoblasts) stimulated myogenic differentiation measured by increased protein expression of myogenin and caveolin 3, which may be of great importance in skeletal muscle disorders [138]. Successively, ovariectomized mice, a model of post-menopausal osteoporosis, treated with LX2931 demonstrated increased cortical thickness and improved mechanical bone parameters (ultimate force and stiffness). Furthermore, reduced osteopenia and enhanced osteoblast activity in vivo (measured by plasma concentrations of procollagen I C-terminal propeptide—PICP) were observed [139]. Less efficacy was noticed in experimental cerebral malaria (ECM). Comparative studies of LX2931 and FTY720 showed that, unlike fingolimod, neither prophylactic nor therapeutic treatment with LX2931 improved the survival rate in mice. However, there was a substantial delay in the onset of symptoms. Additionally, pretreatment with LX2931 significantly decreased IFN-γ levels measured 5 days after infection [140].

## 6. Conclusions

The evidence presented in this review corroborates the significant, albeit ambiguous, function of SPL in the physiology of the nervous system, and in the development and course of neurodegenerative diseases. On the one hand, a large body of evidence documented a direct association between SPL overexpression and disease symptoms, thus emphasizing the protective role of SIP and indicating SPL inhibition as a possible way of restoring disturbed homeostasis. However, a complete lack of SPL or SPL deficiency may also have detrimental effects because the enzyme was demonstrated to regulate such important processes as autophagy, epigenetic modifications (histone acetylation and deacetylation) and synaptic transmission. The possible explanation for this fact can be twofold. The first is associated with an excessive amount of S1P, which disturbs Ca^2+^ homeostasis, eventually promoting cell death. Harmful consequences of SPL deficiency may also be connected to the decreased level of the end products of S1P degradation, HE and P-Etn, which may play a significant role in the regulation of cellular survival and death. Whatever the explanation, both arguments highlight the importance of maintaining a balance between the substrate and products of SPL-catalyzed reaction. It is equally important to further clarify whether the altered SPL activity/expression is a cause or a result of the disease.

This review also summarizes recent studies on the chemicals aiming to modulate SPL expression/activity. To our best knowledge, there is a lack of data on SPL activators, therefore only SPL inhibitors were taken into account. We can distinguish sphingosine analogues, non-lipidic direct inhibitors and functional antagonists among them. Irrespective of the group, the common feature of treatment is the occurrence of peripheral lymphopenia, which is of great importance in autoimmune diseases such as MS or RA. However, a reduction in lymphocyte number may have an adverse effect in pathologies without an inflammatory component. Other deleterious effects (e.g., kidney or heart toxicity) need more clarification, as the literature data are scarce, and the benefit-risk ratio should always be taken into account. Finally, as regards their use in neurodegenerative diseases, it still requires further research concerning the influence on the mechanism underlying neurodegeneration in a particular disease and undesirable peripheral effects.

In summary, SPL plays an important role in neurodegeneration and may become a target of new therapeutic agents developed for the treatment of diseases causing nervous system degeneration.

## Figures and Tables

**Figure 1 ijms-24-06180-f001:**
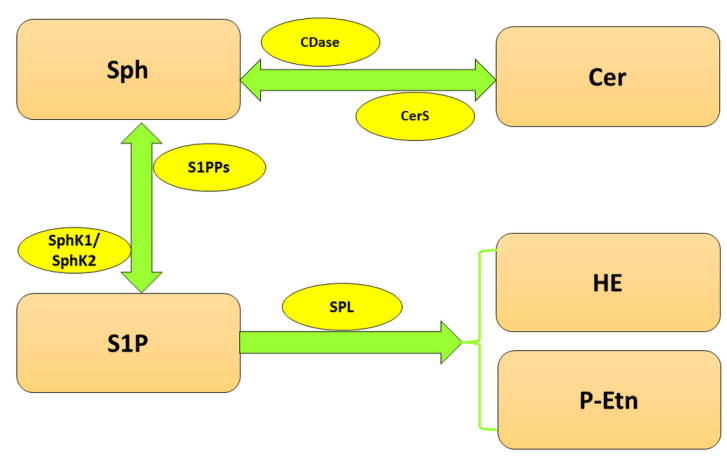
Metabolism of sphingolipids. Abbreviations: CDase—ceramidase; Cer—ceramide; CerS—ceramide synthase; HE—hexadecenal; P-Etn—phosphoethanolamine; Sph—sphingosine; SphK1—sphingosine kinase 1; SphK2—sphingosine kinase 2; S1P—sphingosine-1-phosphate; SPL—sphingosine-1-phosphate lyase; S1PPs—sphingosine-1-phosphate phosphatases.

**Figure 2 ijms-24-06180-f002:**
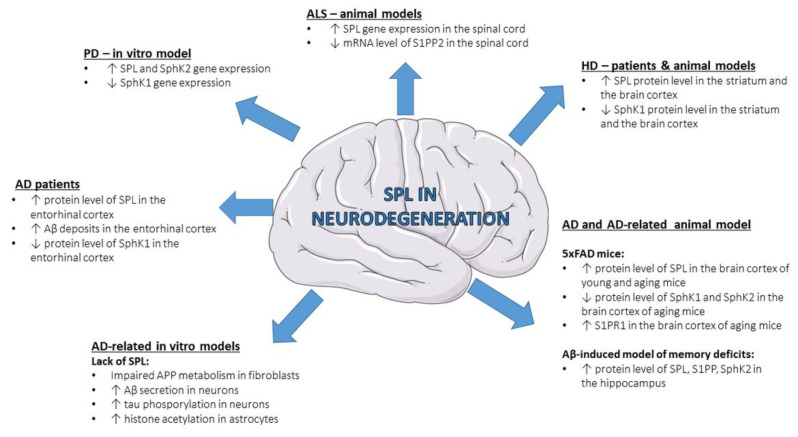
Changes in SPL and other S1P-metabolizing enzymes in neurodegenerative diseases—state-of-the-art data from patients and experimental models. Aβ—amyloid β peptide; AD—Alzheimer’s disease; ALS—amyotrophic lateral sclerosis; HD—Huntington’s disease; S1PP—sphingosine-1-phosphate phosphatase; PD—Parkinson’s disease; S1PP2—sphingosine-1-phosphate phosphatase 2; S1PR1—S1P receptor 1; SphK1—sphingosine kinase 1; SphK2—sphingosine kinase 2; SPL—sphingosine-1-phosphate lyase.

**Table 1 ijms-24-06180-t001:** The summary of the studies on SPL inhibitors.

Inhibitor	Experimental Model	Results	Reference
** *Sphingosine analogues* **			
Fingolimod (FTY720)	HEK293 cell line, 4-week-old FVB mice	↓ SPL activity; no effect on SPL gene expression; no effect on SPL protein level	[115]
	Mouse liver (total tissue and microsomes)	dose-dependent ↓ SPL activity; IC_50_ = 52.4 μM	[116]
	CD68^+^ antigen-presenting cells from human monocytes	↓ hexadecenal (HE) production	[117]
1-desoxysphinganine 1-phosphonate	Rat liver microsomes	inhibitory constant (Ki) = 5 μmol/L ↓ hexadecanal production (end product of dihydrosphingosine-1-phosphate (dhS1P) degradation)	[118]
2-vinyl dihydrosphingosine-1-phosphate (2VS1P)	Rat liver microsomes	IC_50_ = 2.4 μM	[119]
N-[(1R,2S)-2-hydroxy-1-hydroxymethyl-2-(2-tridecyl-1-cyclopentenyl) ethyl] octanamide (GT11)	Murine primary cultured cerebellar neurons	high concentrations of GT11: ↓ SPL activity; accumulation of phosphorylated long-chain bases (S1P, dhS1P); no effect on SPL gene expression	[120]
** *Non-lipidic direct SPL inhibitors* **			
oxopyridylpyrimidine	HepG2 cells	IC_50_ = 2.1 μM; ↑ S1P level	[121]
(R)-6-(4-(4-Benzyl-7-chlorophthalazin- 1-yl)-2-methylpiperazin-1-yl)nicotinonitrile (Compound 31)	(1) purified human SPL (2) HEK293T cells	(1) IC_50_ = 0.214 μM; (2) concentration-dependent ↑ secreted S1P	[122]
	experimental immune encephalomyelitis (EAE) (rat model of MS)	↑ S1P level in lymph nodes, ↑ peripheral lymphopenia, ↓ T cell migration into central nervous system (CNS)	[19]
	EAE (rat model of MS)	platelets activation, skin irritation, kidney toxicity	[123]
	Female and male Sprague-Dawley rats	↑ S1P level in a cardiac tissue, bradycardia	[124]
** *Functional antagonists* **			
4-deoxypyridoxine (DOP)	C57Bl/6 (B6) mice, Ly5.2+ Boy/J mice, RAG2-deficient (129 background) mice	↓ hexadecanal generation in thymus; ↑ lymphoid S1P concentration	[125]
	TNFΔARE mice (TNF-driven model of chronic ileitis)	↓ ileal mRNA level of TNF, IL-6, IL-12, IL-17, IFN-γ; ↓ chronic ileitis, ↓ SPL activity in the ileum	[126]
	Pancreatic islets from Sprague Dawley rats, Pancreatic islets from C57BL/6 mice, mouse insulinoma cell line MIN-6, rat insulinoma cell line INS-1	dose-dependent ↓ level of cleaved caspase 3; ↑ viability	[127]
	10–11-week-old C57BL/6J mice with sepsis induced by intraperitoneal injection of a microbiologically-defined human stool suspension	↓ plasma level of IL-6, TNF-α, monocyte chemoattractant protein (MCP-1) and IL-10; ↑ vascular barrier stability; ↓ immune cell infiltration; ↑ survival; ↑ lymphopenia	[128]
	Primary culture of cerebellar astrocytes from 8-day-old Wistar rats	↑ cellular S1P level; no effect on cell proliferation; no effect on extracellular S1P level	[129]
	mouse embryonic fibroblasts (MEFs)	↑ acetylation of histone 3 at lysine 9 (H3K9)	[99]
2-acetyl-4(5)-(1,2,3,4-tetrahydroxybutyl) imidazole (THI)	C57Bl/6 (B6) mice, Ly5.2+ Boy/J mice, RAG2-deficient (129 background) mice	↓ hexadecanal generation in thymus; ↑ lymphoid S1P concentration	[125]
	BALB/c mice sensitized intraperitoneally with ovalbumin (murine model of allergic rhinitis)	↓ IL-4 level in cervical lymph nodes; ↓ eosinophil and mast cell number in lamina propria of the nasal mucosa	[130]
	SphK1 knockout (SphK1-KO) C57BL/6 mice treated intravenously with KCl (murine model of cardiac arrest)	↑ survival rate; ↑ S1P level in plasma and heart tissue; ↓ plasma level of 22:0 ceramide; ↑ S1PR2 expression	[131]
	Ex vivo murine model of ischemia-reperfusion	↑ cardiac and plasma S1P level; ↓ infarct size,	[132]
	C57BL/6 mice with ligated left anterior descending artery (murine model of myocardial infarction)	↑ plasma level of S1P and ceramide-1-phosphate (C1P); ↑ bone marrow derived stem/progenitor cells (BMSPCs); ↑ expression of genes involved in stem cell survival and myocardial homing	[133]
	Male Wistar rats treated intraperitoneally with streptozotocin (rat model of streptozotocin-induced diabetes)	↑ S1P level in hippocampus, prefrontal cortex, cerebellum and striatum; ↑ ceramide level in hippocampus and cerebellum; ↑ spatial memory	[134]
	(1) R6/2 mice (murine transgenic model of HD) (2) Q128HD-FL transgenic flies (D. melanogaster HD model)	only chronic THI administration: (1) ↓ motor deficits; ↑ prosurvival signaling; ↑ white matter integrity; ↑ proper synaptic activity; (2) ↑ locomotor function; ↑ lifespan	[135]
	mouse primary cortical neurons	↑ Aβ secretion	[90]
(E)-1-(4-((1 R, 2 S, 3 R)-1,2,3,4-Tetrahydroxybutyl)-1 H -imidazol-2-yl)ethanone oxime (LX2931)	129/SvEvBrd x C57BL/6 F1 mice treated with chicken collagen II emulsified in complete Freund’s adjuvant (murine model of collagen-induced arthritis)	↓ circulating lymphocytes; ↓ inflammation, erosion, synovial hyperplasia and exudate formation	[136]
	Cftr^tm1EUR^ F508del CFTR mice (murine model of cystic fibrosis)	↑ S1P level in lungs; ↓ level of IFN-γ, IL-12, IL-10, keratinocyte-derived cytokines (KC) in lungs; ↓ production of an inflammation-responsive mucin Muc5AC	[137]
	C2C12 cell line (murine myoblasts)	↑ S1P level; ↑ protein level of myogenin and caveolin 3	[138]
	Ovariectomized 12-week- old C57Bl6J mice (murine model of postmenopausal osteoporosis)	↓ osteopenia; ↑ osteoblast activity; ↑ cortical thickness; ↑ mechanical bone parameters (ultimate force and stiffness)	[139]
	6- to 8-week-old female C57BL/6 mice infected with *Plasmodium berghei* ANKA strain (murine model of cerebral malaria)	delay of symptoms onset; ↓ IFN-γ plasma level 5 days after infection	[140]

Abbreviations: C1P—ceramide-1-phosphate; CNS—central nervous system; dhS1P—dihydrosphingosine-1-phosphate; EAE—experimental immune encephalomyelitis; HE—hexadecenal; IC_50_—half maximal inhibitory concentration; KC—keratinocyte-derived cytokines; Ki—inhibitory constant; MCP-1—monocyte chemoattractant protein 1; ↓—decrease; ↑— increase.

## Data Availability

No new data were created or analyzed in this study. Data sharing is not applicable to this article.

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
