# Peer review of "Recent Insight into the Role of Sphingosine-1-Phosphate Lyase in Neurodegeneration"

_ijms, 2023, doi:10.3390/ijms24076180_

Round 1
Reviewer 1 Report
This is a comprehensive review of an enzyme (SPL) located at a strategic position in the lipid metabolism. It also includes a good deal of useful pharmacological information related to the enzyme (Table 1). The review is well-planned and structured, the narrative is fluent and there are no major language issues. I have a few queries as listed below:
# 1 Line 27: “On the one hand.. “ should be logically followed by "on the other hand".
# 2 Line 52: Can it be specified which ER (smooth or rough)? Do their respective membranes differ with respect to the content (and/or the role) of SPL? It might be tactically wise to introduce the position of the active site (facing cytosol) at this stage and then continue at line 60, to emphasize the point.
# 3 Line 66: Which of the cells of the epithelium?
# 4 Line 67: Explain why SPL1 is so important in neurodegeneration when it is "barely expressed" in brain? Is this actually a correct statement (https://www.proteinatlas.org/ENSG00000166224-SGPL1/tissue) or am I missing something somewhere? See also line 118.
# 5 Lines 101/102: Nonenzymic reactions in biological systems tend to be intriguing. Perhaps the authors might comment on the potential significance of HE formed in this way. Could it perhaps be increased in the presence of higher-than-normal free radical concentrations, or, perhaps, in the presence of particular free radical species?
# 6 Line 139: You may wish to specify from which intracellular stores is Ca2+ released and by what mechanism, if known.
# 7 Line 160: Disturbed synaptic neurotransmission could be a mechanism very significantly contributing to neurodegeneration. It could be worthwhile to mention whether the deficiencies in the presynaptic proteins selectively influence a particular type of synapses – inhibitory v. excitatory? How does the lack of SPL cause the protein deficiencies?
# 8 Line 163: Is a particular type of vesicles preferentially impacted (this could give a clue as to whether the two major types, glutamatergic v. GABAergic synapses were differentially affected, see previous comment)?
# 9 Line 272: The effect pf MPP+ is intriguing – any ideas on the molecular mechanism?
# 10 Line 396: Could the actions of fingolimod be mediated by its effects on CD36 or PPAR-gamma (DOI 10.3389/fnmol.2022.1077381)? Which effects come first as the concentration of fingolimod increase, those on SPL or those on PPAR-gamma and CD36 (I am aware of fingolimod being a “prodrug”)? Note that changes in CD36 have recently been implicated in the mechanisms of Alzheimer’s disease (https://doi.org/10.1038/s41598-022-15299-z)
# 11 Authors state that one of the aims of the review is to look at possible role of SPL in neurodegeneration and they mention SPL as a possible therapeutic target. This obviously implies that at least some of the compounds listed in Table 1 could become useful drugs in neurodegenerative diseases, whether it be as actual therapeutics or tools to create suitable animal models. Is there anything known on how they penetrate the blood brain barrier?
# 12 I note that SPL is encoded by a single non-multiplicated gene in all organisms where it has been studied to date. Does this imply that deleterious mutations would be particularly damaging to the carrier? How is the structure of the enzyme (amino acid sequences) conserved across the species?
# 13 What is the EC number of SPL activity? It would be neat and useful to have it somewhere in Introduction.
Author Response
We would like to thank you for your time and helpful comments and suggestions. You can find our answers to your questions/suggestions below in attached file.

Reviewer 2 Report
The review article Recent insight on the role of sphingosine-1-phosphate lyase in neurodegeneration by the authors Iga Wieczorek and Robert P. Strosznajder studies rather interesting, but unfairly neglected and obscure role of enzyme sphingosine-1-phosphate lyase in neurodegenerative diseases. This excellent review summarizes the current findings concerning the role of SPL in the nervous system with an emphasis on neurodegeneration. More interestingly, it briefly discusses pharmacological compounds inhibiting its activity providing a directive to the researchers worldwide when it comes to potential novel therapeutic strategies in neurodegeneration.
The review is written in a concise, but a comprehensive manner successfully presenting the data relevant to the topic.
However, the only suggestion to authors is to shorten the Section SPL in the nervous system or even completely omit it. Given the primary goal of this review, a summary of SPL role in neurodegenerative disorders/diseases (Alzheimer’s disease, Parkinson’s disease, Huntington’s disease, amyotrophic lateral sclerosis, and spinal muscular atrophy), data presented in this section seems a little excessive.
Author Response
We would like to thank you for your time and helpful comment. You can find answer to your suggestion below.
However, the only suggestion to authors is to shorten the Section SPL in the nervous system or even completely omit it. Given the primary goal of this review, a summary of SPL role in neurodegenerative disorders/diseases (Alzheimer’s disease, Parkinson’s disease, Huntington’s disease, amyotrophic lateral sclerosis, and spinal muscular atrophy), data presented in this section seems a little excessive.
The information about the role of SPL in the nervous system is important to understand the significance of this enzyme in neurodegeneration; hence in our opinion, this section is necessary. However, according to the reviewer’s suggestion, the last paragraph of this section (the role of SPL in autophagy) was slightly shortened because we agree that some information in this paragraph exceeds the topic of this publication. See lines 175-190 in attached file.
We thank you again, and we hope our answers will satisfy you.

Reviewer 3 Report
Sphingosine-1-phosphate lyase (SPL) is a pyridoxal 5’-phosphate-dependent enzyme involved in the irreversible degradation of sphingosine-1-phosphate (S1P which modulates cell proliferation, migration, differentiation, and survival along with mitochondrial functioning and gene expression. SPL decreases the available pool of S1P in the cell by generating potentially active metabolites, hexadecenal and phosphoethanolamine. The increased expression or/and activity of SPL was demonstrated in several pathological states. Loss-of-function mutations in SPL encoding gene are a cause of severe developmental impairments. Current findings concerning the role of SPL on neurodegeneration are also summarized. In addition, Authors briefly discuss pharmacological compounds that inhibit SPL activity.
English style and grammar could be improved.
Author Response
We would like to thank you for your time and helpful comment regarding English style and grammar. The publication has been carefully checked in terms of style, spelling and grammar (we have also consulted linguistic issues with a native speaker).
The entire publication has been corrected with the utmost care regarding the language.
Please see attached manuscript after english correction

Round 2
Reviewer 1 Report
Authors did react to some of my suggestions and improved the text here and there but merely acknowledged and ignored others comments. This applies also to their "brief" discussion of pharmacological compounds acting on SPL. In the case of fingolimod authors appear to accept that the actions on PPAR-gamma and CD36 could occur before those on SPL. This could provide alternative explanations of its anti inflammatory actions I was hoping that this would included in some way in the text. There are around forty references to fingolimod/FTY720 but without a greater emphasis on non-SPL targets and alternative mechanisms (as I pointed out in my first report) and possible side effects, the review lacks proper balance and could be seen as too biased by more objective and/or knowledgeable readers. At the very least, one or two recent reviews should be cited (e.g. J Chun et a doi.org/10.1146/annurev-pharmtox-010818-021358 or Constantinescu et al doi.org/10.1080/17425255.2022.2138330).
Author Response
Authors did react to some of my suggestions and improved the text here and there but merely acknowledged and ignored others comments. This applies also to their "brief" discussion of pharmacological compounds acting on SPL. In the case of fingolimod authors appear to accept that the actions on PPAR-gamma and CD36 could occur before those on SPL. This could provide alternative explanations of its anti-inflammatory actions I was hoping that this would be included in some way in the text. There are around forty references to fingolimod/FTY720 but without a greater emphasis on non-SPL targets and alternative mechanisms (as I pointed out in my first report) and possible side effects, the review lacks proper balance and could be seen as too biased by more objective and/or knowledgeable readers. At the very least, one or two recent reviews should be cited (e.g. J Chun et a doi.org/10.1146/annurev-pharmtox-010818-021358 or Constantinescu et al doi.org/10.1080/17425255.2022.2138330).
Thank you for your comment. You can find answers to your questions/suggestions below.
“In the case of fingolimod authors appear to accept that the actions on PPAR-gamma and CD36 could occur before those on SPL.”
This claim is only a hypothesis (there is no research concerning the effect of FTY720 on both PPAR-γ and SPL simultaneously), and it is also possible that the actions on SPL could occur before those on PPAR-γ and CD36. In accordance with the topic of the chapter, we focused on the action of fingolimod on SPL. We mentioned, of course, the primary mechanism of action of this drug as some introduction (although FTY720 can also act on other enzymes involved in sphingolipid metabolism [doi.org/10.3389/fncel.2014.00283] or, as you mentioned, on PPAR-γ, but the description of all possible targets of FTY720 exceeds this article). Readers more interested in the FTY720-specific mechanisms can find review articles focused on that issue.
“This could provide alternative explanations of its anti-inflammatory actions I was hoping that this would be included in some way in the text.”
Comment No. 10 from the first round of reviews (regarding the link between FTY720 and PPAR-γ) didn’t suggest that this information should be included in the text.
“At the very least, one or two recent reviews should be cited (e.g. J Chun et a doi.org/10.1146/annurev-pharmtox-010818-021358 or Constantinescu et al doi.org/10.1080/17425255.2022.2138330).”
According to the reviewer’s suggestions, the list of references has been supplemented with mentioned papers.
The paper of Chun et al.: line 356, reference 143
The paper of Constantinescu et al.: line 368, reference 155
